# Using Computational Fluid Dynamics to Evaluate High Tunnel Roof Vent Designs

**David C. Lewus** [1,*] **and Arend Jan Both** [2]

1 Department of Plant Biology, Rutgers University, New Brunswick, NJ 08901, USA
2 Department of Environmental Sciences, Rutgers University, New Brunswick, NJ 08901, USA
* Correspondence: david.lewus@rutgers.edu

**Abstract:** Freestanding high tunnels are cost-effective, plastic film-covered growing structures that use very little to no modern environmental control technology. Natural ventilation is used to control temperature and humidity. Typically, ventilation openings are created along the sides by manually rolling up a section of the plastic film cover. While common on greenhouses, roof vents are not typically part of high tunnel designs used in the United States. This paper focuses on high tunnel ventilation during the summer, when maximizing the air exchange rate results in a low differential between inside and outside air temperatures. Computational fluid dynamics (CFD) simulations were used to evaluate the effects of several roof vent designs on the air flow rate through the high tunnel and the inside air temperature. The CFD models were developed and validated using environmental data collected at the Pennsylvania State University High Tunnel Research and Education Facility (Rock Springs, PA, USA). Five ventilation designs were simulated using a commercial CFD software package that was augmented with a radiation and crop architecture model. A root mean square error of 0.87 °C was found between the measured and simulated high tunnel temperatures (n = 144). The designs with roof vents were found to increase mass-based ventilation rates through the high tunnel by 20% to 78%. However, they did not lower inside air temperature more than 0.1 °C compared to the traditional design with roll-up side vents only. Additional research is needed to evaluate whether the control of other environmental parameters and weather conditions warrants the use of high tunnel roof vents, especially for humidity control and the combination of high temperature with low wind speed conditions.

**Keywords:** air exchange; CFD; simulation; temperature control; ventilation

## 1. Introduction

The use of high tunnels for season extension offers an important economic opportunity for North American farmers, especially those in the temperate regions of the United States. High value crops such as *Rubus idaeus* (red raspberry) or *Rubus occidentalis* (black raspberry) are of particular interest because of the demand and market price in the United States. During the off-season, between October and May, hundreds of millions of kilograms of fresh raspberries are imported into the United States, primarily from Mexico [1]. Import amounts have been steadily increasing over the past decade, indicating a strong market interest for fresh raspberries during the off-season. High tunnels can extend seasonal production by several weeks at the beginning and end of the growing season across the temperate regions of the United States [2,3]. Furthermore, high tunnels can cost four times less to install compared to traditional plastic greenhouses [4]. This cost difference is mainly because high tunnels are non-permanent structures (i.e., no poured concrete foundation) and do not have sophisticated environmental control systems (e.g., heating system, evaporative cooling, mechanical ventilation) [3].

While freestanding high tunnels are effective for season extension, growers typically only use high tunnel roll-up vents to achieve environmental control. These vents are

commonly placed on the side walls. Natural airflow caused by thermal buoyancy and the wind effect passively ventilates these structures, mixing outside air with the air inside the structure. This is necessary to remove heat trapped from incoming solar radiation and to remove water vapor that is generated by evapotranspiration. Without natural ventilation during the summer, when the sun is up, temperatures can rise to levels intolerable to raspberry plants (and most other commercial crops); moreover, throughout the entire day, humidity can rise to levels that can greatly increase the risk of disease incidence and limit the plants ability to transpire. Such environmental conditions that can occur without adequate ventilation will ultimately lead to reduced yields.

Previous research has been conducted to investigate the ventilation efficiency in naturally ventilated plastic-covered single-bay greenhouses, which are similar in design to freestanding high tunnels. Much of this work has been done using computational fluid dynamics (CFD) simulations because of the cost and complexity of alternate methods such as wind tunnel experiments or tracer gas tests [5]. Additionally, availability of sufficient computing power for CFD modeling has become more affordable and easier to access for researchers. Hong et al. [6] described methods for characterizing ventilation rates in a variety of naturally ventilated multi-span greenhouses using volumetric flow rates and tracer gas decay methods, and reported how these could be applied to CFD modeling. Boulard and Wang [7] implemented a porous media model to characterize the drag effects of a crop. Shklyar and Arbel [8] developed a model to analyze isothermal flow patterns produced under turbulent airflow.

Bartzanas et al. [9] investigated the effects of ventilation configuration of a greenhouse on airflow and temperature distribution. Bournet et al. [10] simulated the effect of radiative heat exchange within a high tunnel and its effects on the tunnel temperature. Rasheed et al. [11] simulated the effects of solar radiation and a variety of greenhouse vent designs on the inside-outside temperature differential and air exchange rate. All these studies used steady-state simulations. Few studies have used transient simulations to assess the greenhouse conditions over the course of a day. Bournet et al. [12] simulated the environmental dynamics in a greenhouse with a crop model and a radiation model. Bouhoun Ali et al. [13,14] used similar models to simulate crop water use, leaf temperatures, and relative humidity. All of these greenhouse condition studies used 2-D CFD models.

Overall, these studies have created a strong scientific basis for CFD investigations into the dynamics of natural ventilation in plant production systems. Nonetheless, there has been little research on the effects of varying vent designs on the evolution of internal conditions of a high tunnel used for raspberry production, especially with a 3-D CFD model.

The aim of this study was to use CFD techniques to evaluate the ventilation performance in freestanding high tunnels used for raspberry production over the course of an entire day. Additionally, this study aimed to determine if a roof vent improves high tunnel summer ventilation. The ventilation performance was simulated, and modeling results were validated using environmental data collected in a high tunnel with no roof vents located at the Pennsylvania State University High Tunnel Research and Education Facility (Rock Springs, PA, USA).

## 2. Materials and Methods

In this study, the ventilation performance of five high tunnel vent designs was simulated. The mass-based ventilation rate and inside-outside temperature differential were used to describe the ventilation performance of the high tunnels. A commercially available CFD software package (ANSYS Fluent) was used to simulate the ventilation performance of the five designs [15]. All simulation results were compiled and analyzed using Microsoft Excel.

### 2.1. High Tunnels

A single freestanding high tunnel at the Pennsylvania State University High Tunnel Research and Education Facility (Rock Springs, PA, USA) was used for this study (located

at 40.711° N, 77.945° W and at 369 m elevation above sea level) and environmental data were collected from 21 August 2018 through 15 November 2019. The dimensions of this tunnel are shown in Figure 1. The tunnel was oriented approximately 10 degrees west of north–south so that wind coming from the prevailing wind direction (west to east) would strike perpendicularly to the windward side vent. The tunnel was outfitted with two roll-up side vents and with a continuous rack-and-pinion roof vent along the east (leeward) side. This roof vent remained closed for the data collection period and was used to help design the roof vent configurations evaluated in this study. The tunnel was covered with a single layer of 0.152 mm thick transparent polyethylene greenhouse film (Sunsaver, Ginegar Plastic Products Ltd., Kibbutz Ginegar, Israel). Environmental data collected inside the high tunnel included: temperature, relative humidity, soil heat flux (at 0.1 m below the soil), wind speed and direction, photosynthetically active radiation (PAR), and solar radiation. A weather station was placed nearby and outside the high tunnel, and captured solar radiation, wind speed and direction, temperature, and relative humidity. The sensors used are listed in Table 1. The thermocouples were placed inside aspirated boxes to avoid measurement errors due to solar radiation directly heating the sensor. All sensors, except for the thermocouple probes, were calibrated by the manufacturer. The thermocouple probes were calibrated using 0 °C and 100 °C water.

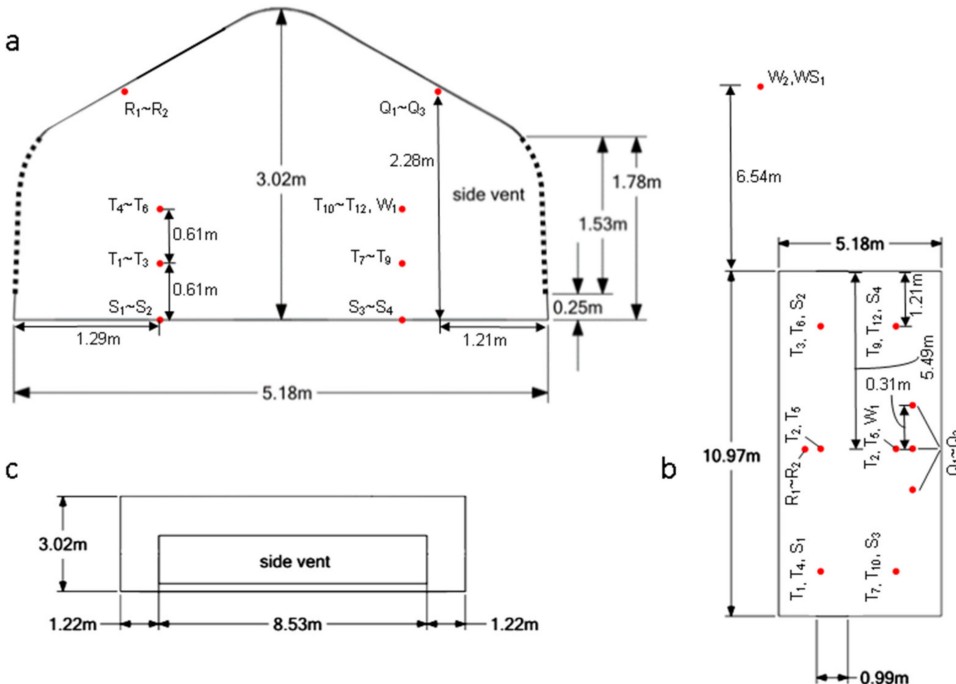

**Figure 1.** Dimensions of and sensor locations inside the experimental high tunnel: (**a**) end, (**b**) top, and (**c**) windward side view. T = air temperature and relative humidity; S = soil heat flux and soil temperature; R = solar radiation (LI200X); Q = PAR; W = wind speed and direction ($W_1$ = CSAT3B, $W_2$ = 034B-L, WS = Weather Station).

**Table 1.** Sensors used to collect environmental data inside and outside the high tunnel.

| Parameter | Sensor | Range | Accuracy | Manufacturer |
|---|---|---|---|---|
| Air and Soil Temperature | PFA-Insulated 24-AWG Type-T Thermocouple | −250 °C to 350 °C | Greater of ± 1.0 °C or ± 0.75% | OMEGA Engineering |
| Relative Humidity | HMP60-L | 0 to 100% RH | ± 3% [1] | Campbell Scientific |
| Soil Heat Flux, 0.1 m below soil surface | HFP01-L | ± 2000 W m$^{-2}$ | −15% to 5% | Hukseflux |
| Wind Speed/Direction | CSAT3B | 0 to 65 m s$^{-1}$ 0 to 360° | ± 3% [2] | Campbell Scientific |

**Table 1.** *Cont.*

| Parameter | Sensor | Range | Accuracy | Manufacturer |
|---|---|---|---|---|
| Wind Speed/Direction | 034B-L | 0 to 75 m s$^{-1}$<br>0 to 360° | 0.1 m s$^{-1}$<br>± 4° | Campbell Scientific |
| PAR | Line Quantum Sensor [3] | 0 to 2500 µmol m$^{-2}$ s$^{-1}$ | ± 5% | Apogee Instruments Inc. |
| Solar Radiation | CS300 | 0 to 2000 W m$^{-2}$ | ± 5% | Campbell Scientific |
| Solar Radiation | LI200X | 0 to 3000 W m$^{-2}$ | ± 5% | LI-COR Biosciences |
|  |  | 0 to 1750 W m$^{-2}$ | ± 5% |  |
| Weather Station [4] | CLIMAVUE50 | 0 to 30 m s$^{-1}$<br>0 to 359°<br>−50 °C to 60 °C<br>0 to 100% RH | 0.3 m s$^{-1}$<br>± 5°<br>± 0.6 °C<br>± 3% RH | Campbell Scientific |

[1] at 0° to 40 °C and 0 to 90% RH, which are the ranges experienced for the majority of the dataset. [2] wind vector within ± 10° of horizontal, which is the range experienced for the majority of the dataset. [3] An older model line quantum sensor with six sensors. [4] The weather station measured solar radiation, wind speed and direction, temperature, and relative humidity.

## 2.2. CFD Computational Domain

CFD models were developed using Fluent software [15]. Using this software, a three-dimensional high tunnel model was created and discretized into a mesh over a computational domain. Parameters needed to develop this mesh and domain were adapted from Kim et al. [16]. The first cell height was set to 0.003 m ($y^+$ = 5) to allow accurate near-wall modeling. Mesh inflation was used at the first five layers of cells to avoid highly skewed cells due to this low $y^+$. In order to adequately simulate the effects of the atmospheric boundary layer and surrounding wind conditions, a sufficiently sized computational domain must be developed. At the same time, this domain should be as small as possible to still capture relevant effects while minimizing computational resources. The computational domain was generated by extending a length of three times the maximum height of the high tunnel (3H) upstream, 15H downstream, 5H on either side of the high tunnel, and 5H above the tunnel. This domain size is shown in Figure 2. A mesh face size of 0.08 m was used for the high tunnel volume. The surface mesh was generated using the curvature and proximity function, and the volume mesh was filled with a polyhedral shape. The total number of cells was approximately 600,000. The high tunnel with roll-up side vents only was modeled using the dimensions shown in Figure 1. It was assumed to be airtight except at the vents. Rows of raspberry plants were modeled as blocks of porous media, as described by Boulard and Wang [7]. The plants were simulated using three rows of rectangular prisms, and their dimensions are shown in Figure 3.

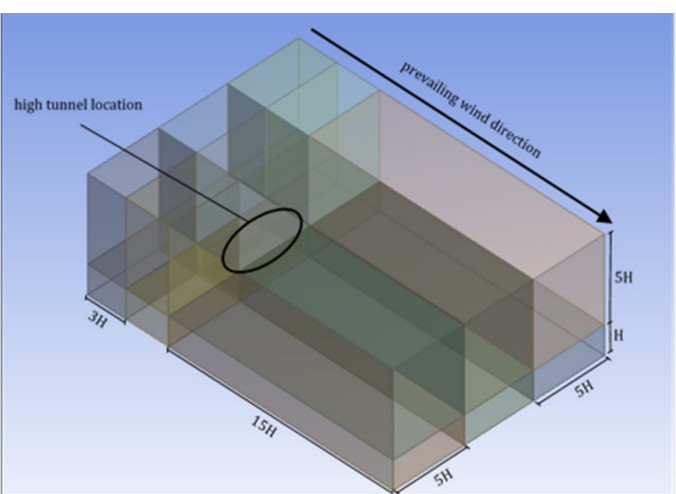

**Figure 2.** Computational domain used for the CFD simulations.

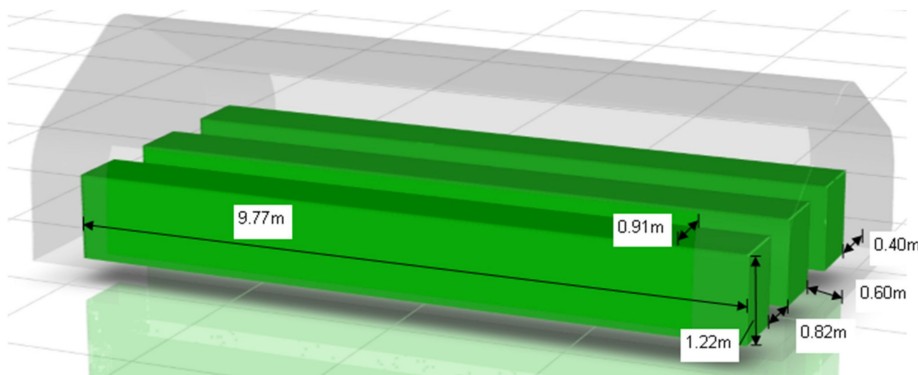

**Figure 3.** Dimensions of three simulated plant rows located inside a freestanding high tunnel.

### 2.3. CFD Boundary Conditions

The CFD software solved the Navier–Stokes equations for mass, energy, and momentum balances to calculate the air flow and temperature. The standard k-ε turbulence model was used as it has shown good accuracy and low computational load [17]. The discrete ordinates (DO) radiation model was used with two bands to discretize radiation into shortwave (0–0.7 μm) and longwave (0.7–2.5 μm) radiation. This radiation model was chosen because it has shown good accuracy and moderate computational cost in related studies [10]. Fluent's solar calculator was used to determine the solar ray angle, and radiation was applied parallel to this beam. Figure 4 shows the five different configurations of the vents used in this study. These designs mimic some of the designs commonly used on plastic greenhouses. A transient simulation was run to assess the dynamics of the air flow and temperature over the course of an entire day (i.e., 24 h from 0:00 to 24:00). The environmental input parameters and initial conditions (Table 2) for air temperature, solar radiation, wind speed, and wind direction were obtained from the experimental high tunnel on 13 July 2019 (Figure 5) and the sensor locations are shown in Figure 1. This day was used because it was warm (average of 21.9 °C), there was almost no cloud cover, and the wind direction was mostly perpendicular to the vents; conditions that have been shown to lead to the highest air speed through a tunnel [18]. These conditions may limit the broader applicability of the study results since there can be considerable effects from cloud cover on radiation and wind speed, and from wind direction on air flow. While humidity data were collected in the experimental high tunnel, these data were not used to validate the simulations. Table 3 shows the material properties used in the simulation.

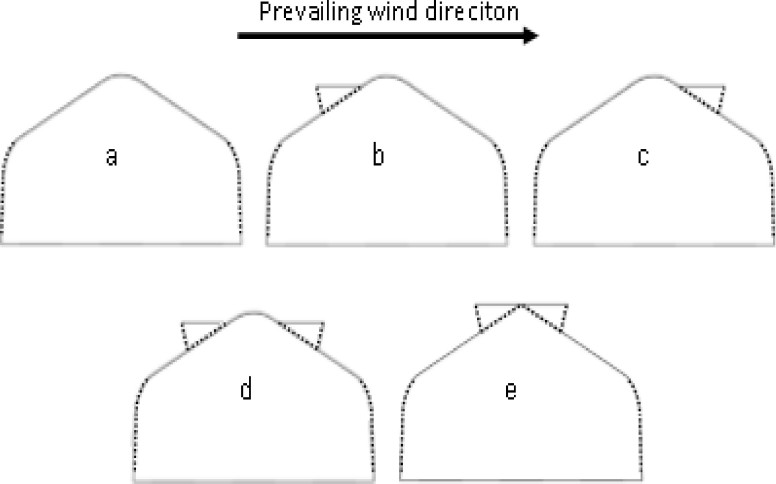

**Figure 4.** High tunnel vent configurations: (**a**) roll-up side vents only; (**b**) with added windward roof vent; (**c**) with added leeward roof vent; (**d**) with added windward and leeward roof vents; and (**e**) with added center ridge vents.

**Table 2.** Input values used as initial conditions at 12:00 am (at midnight), measured on 13 July 2019.

| Parameter | Value |
|---|---|
| Air speed at 2.5 m, m s$^{-1}$ | 0.1 |
| Air temperature, °C | 15.8 |
| Solar radiation, W m$^{-2}$ | 0 |

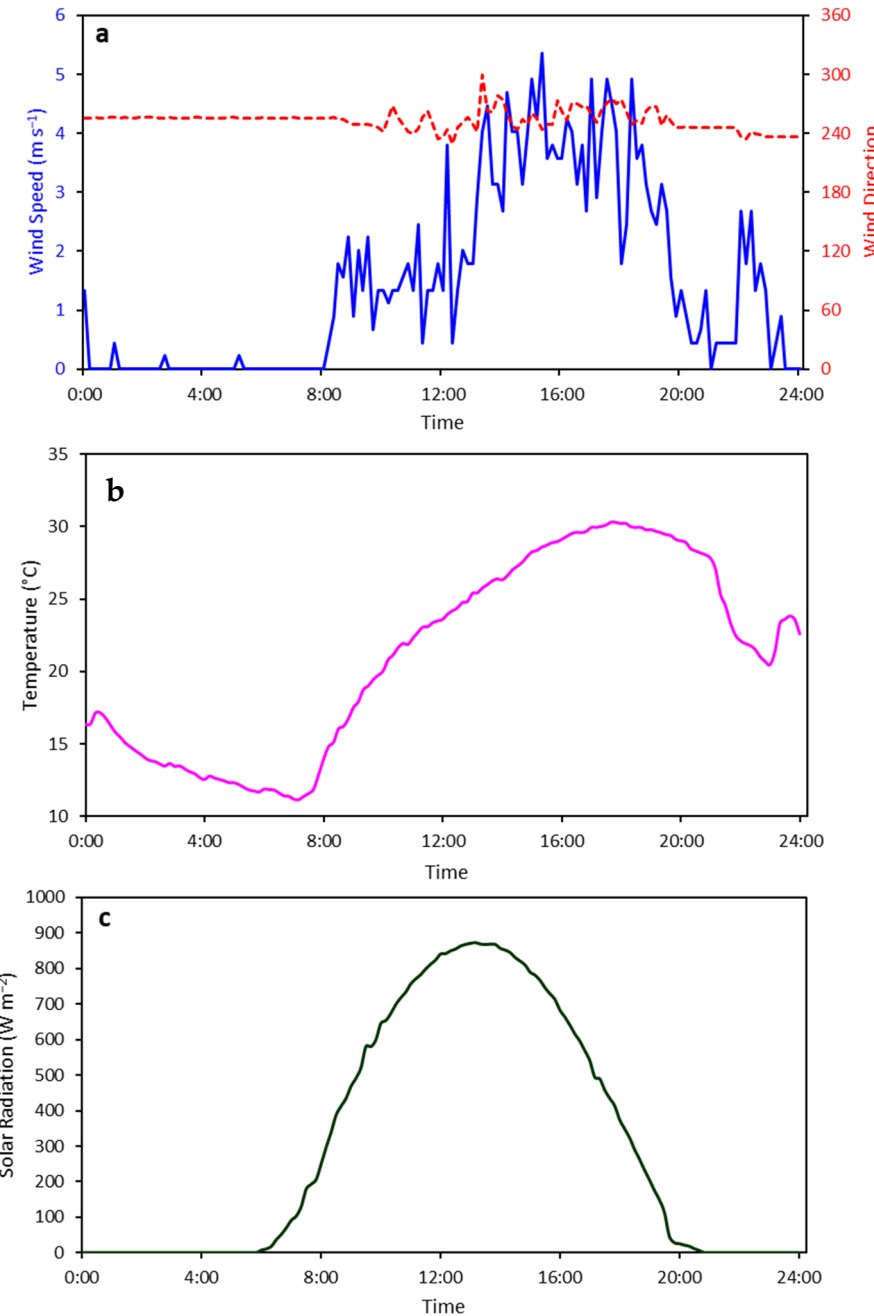

**Figure 5.** Measured outside wind speed and wind direction at 2.5 m (**a**), measured outside temperature (**b**), and measured solar radiation (**c**) on 13 July 2019. Note that the experimental high tunnel was oriented in such a way that a wind direction of approximately 260° would be perpendicular to the roll-up side vents and entering the windward vent first.

**Table 3.** Material properties used during the CFD simulations.

| Material | Parameter | Value |
| --- | --- | --- |
| Air [15] | Specific heat, J kg$^{-1}$ K$^{-1}$ | 1006.43 |
| | Density, kg m$^{-3}$ | 1.225 |
| | Thermal conductivity, W m$^{-1}$ K$^{-1}$ | 0.0242 |
| | Viscosity, kg m$^{-1}$ s$^{-1}$ | $1.7894 \times 10^{-5}$ |
| Soil [19–21] | Specific heat, J kg$^{-1}$ K$^{-1}$ | 800 |
| | Density, kg m$^{-3}$ | 1300 |
| | Thermal conductivity, W m$^{-1}$ K$^{-1}$ | 1 |
| LDPE plastic cover [22] | Specific heat, J kg$^{-1}$ K$^{-1}$ | 1900 |
| | Density, kg m$^{-3}$ | 920 |
| | Thermal conductivity, W m$^{-1}$ K$^{-1}$ | 0.33 |
| | Absorption coefficient, m$^{-1}$ | 1.0385 |
| | Refractive Index | 1.5 |
| Canopy [23,24] | Specific heat, J kg$^{-1}$ K$^{-1}$ | 2310 |
| | Density, kg m$^{-3}$ | 700 |
| | Thermal conductivity, W m$^{-1}$ K$^{-1}$ | 0.173 |
| | Absorption coefficient, m$^{-1}$ | 1.923 |
| | Refractive Index | 2.77 |

*2.4. Model Validation*

The model was validated using temperature data collected inside the experimental high tunnel with fully opened roll-up side vents and populated with raspberry plants. The tunnel temperature measured throughout the day was compared to the simulated temperature for the case with fully opened roll-up side vents (Figure 4a).

### 3. Results

As described above, the model validation compared experimentally recorded temperatures to simulated temperatures for the design with only roll-up side vents. This comparison can be seen in Figure 6. Simulated data were collected for three cross sections in the modeled high tunnel. A root mean square error (RMSE) of 0.87 °C was determined (n = 144), showing good agreement between experimental and simulated results since this error is within the measurement error of the temperature sensors used. The maximum and average differences between the experimental and simulated temperatures were 14.9% and 3.6%, respectively. This large maximum difference happened at night where the model may not have been able to adequately characterize the conditions present at that time of low convection and infrared radiative emission. Additionally, the experimental data only measured air temperature outside of the canopy, while the simulated data also included the canopy temperature. The crop may have acted as a heat sink in such a way that the heat was not evenly distributed in a low wind environment at night.

The total mass-based ventilation rates through the leeward vent/vents (both roll-up side and roof vent if applicable) over the course of the day are shown in Figure 7, and average values are shown in Table 4. The ventilation rate at the leeward roll-up side vent over the course of the day is shown in Figure 8. This metric describes how much air was exhausted through the leeward vent/vents. A higher value would imply a greater air exhaust rate, but does not necessarily imply a lower inside-outside temperature differential. Figures 7 and 8 illustrate a consequence of leeward roof vents that is not obvious by just looking at the total ventilation rate. While higher ventilation rates at all the leeward vents were observed for high tunnel designs with leeward roof vents (shown in Figure 4c–e), these roof vent designs did not improve the ventilation rate at the leeward roll-up side vent. This is important because flow through this vent typically helps improve ventilation of an aerodynamically resistant canopy.

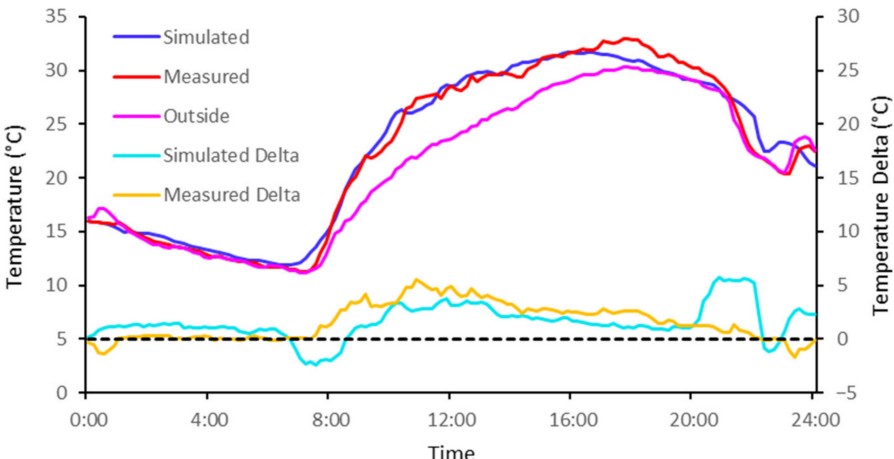

**Figure 6.** Simulated versus measured temperatures inside a high tunnel on 13 July 2019. The simulated and measured temperatures were compared to the measured outside temperature and expressed as differences (delta).

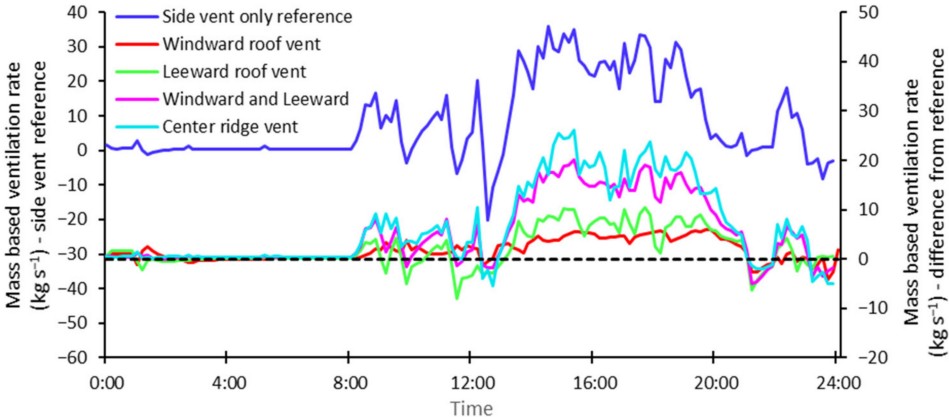

**Figure 7.** Mass-based ventilation rate at the leeward vent/vents for the different roof vent designs evaluated. The primary y-axis represents the actual air flow rates for the side vent only design as a reference. The right axis represents the air flow rates at the leeward vent/vents of the other designs subtracted by the side vent only reference (positive values mean higher air flow rates relative to the reference case).

**Table 4.** Simulated average mass-based ventilation rates through the leeward vent(s) depending on the configuration of the roof vent(s). Average outside wind speed at 2.5 m: $1.5 \text{ m s}^{-1}$. Wind direction: 96.4° (relative to side vents). High tunnel dimension: 11 m × 5.2 m × 3 m (length × width × height).

| Roof Vent Design | Average Mass-Based Ventilation Rate through the Combined Leeward Vents, $\text{kg s}^{-1}$ | Average Mass-Based Ventilation Rate through the Leeward Side Vent only, $\text{kg s}^{-1}$ |
|---|---|---|
| Roll-up side vents only | 9.3 | 9.3 |
| Added windward roof vent | 11.1 | 11.1 |
| Added leeward roof vent | 11.6 | 6.0 |
| Added windward and leeward roof vents | 15.0 | 7.9 |
| Added center ridge vents | 16.5 | 9.2 |

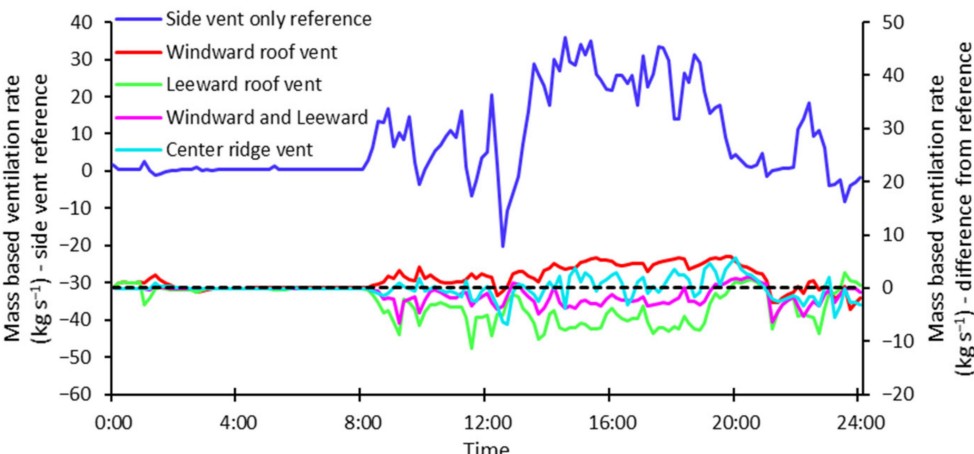

**Figure 8.** Mass-based ventilation rate at the leeward roll-up side vent for the different roof vent designs evaluated. The primary y-axis represents the actual air flow rates for the side vent only design as a reference. The right axis represents the air flow rates at the leeward roll-up side vents of the other designs subtracted by the side vent only reference (positive values mean higher air flow rates relative to the reference case).

Figure 9 shows velocity vectors for three cross sections of the five tunnel designs at 15:00 h when the air flow was the highest. These vectors illustrate airflow patterns through each tunnel design and raspberry canopy. For all roof vent designs, the airflow passes through the windward vent/vents to the leeward vent/vents, showing that ventilation at this time is mostly driven by convection and not by buoyancy. Furthermore, the air flows roughly perpendicular to the vents. There are critical differences between all of these designs that explain the differences in ventilation rates shown in Table 4. In Figure 9a (roll-up side vents only), air passes over and through the plants, with little air exchange near the peak of the tunnel. In Figure 9b (windward roof vent), a similar airflow pattern is seen, except the magnitude of the velocity vectors are greater near the peak of the tunnel. In Figure 9c (leeward roof vent), air is split between the leeward side vent and the leeward roof vent, which causes less air to pass through the plants compared to the conditions shown in Figure 9a,b. This pattern develops as soon as the air enters the windward side vent, evidenced by a series of vectors pointing up towards the leeward roof vent. A pattern similar to Figure 9c is seen in Figure 9d (windward and leeward roof vent) and 9e (ridge vent), except in Figure 9e the vectors have greater magnitude throughout the tunnel. This makes up for the air diverted towards the leeward roof vent and allows for more air to pass through the plants and the leeward side vent. The simulated internal temperature over the course of the day is shown in Figure 10, and average values are shown in Table 5. These simulated temperatures were calculated as averages over the respective cross sections depicted in Figure 9. The temperature differences are very similar and would not likely have a differential impact on the crop.

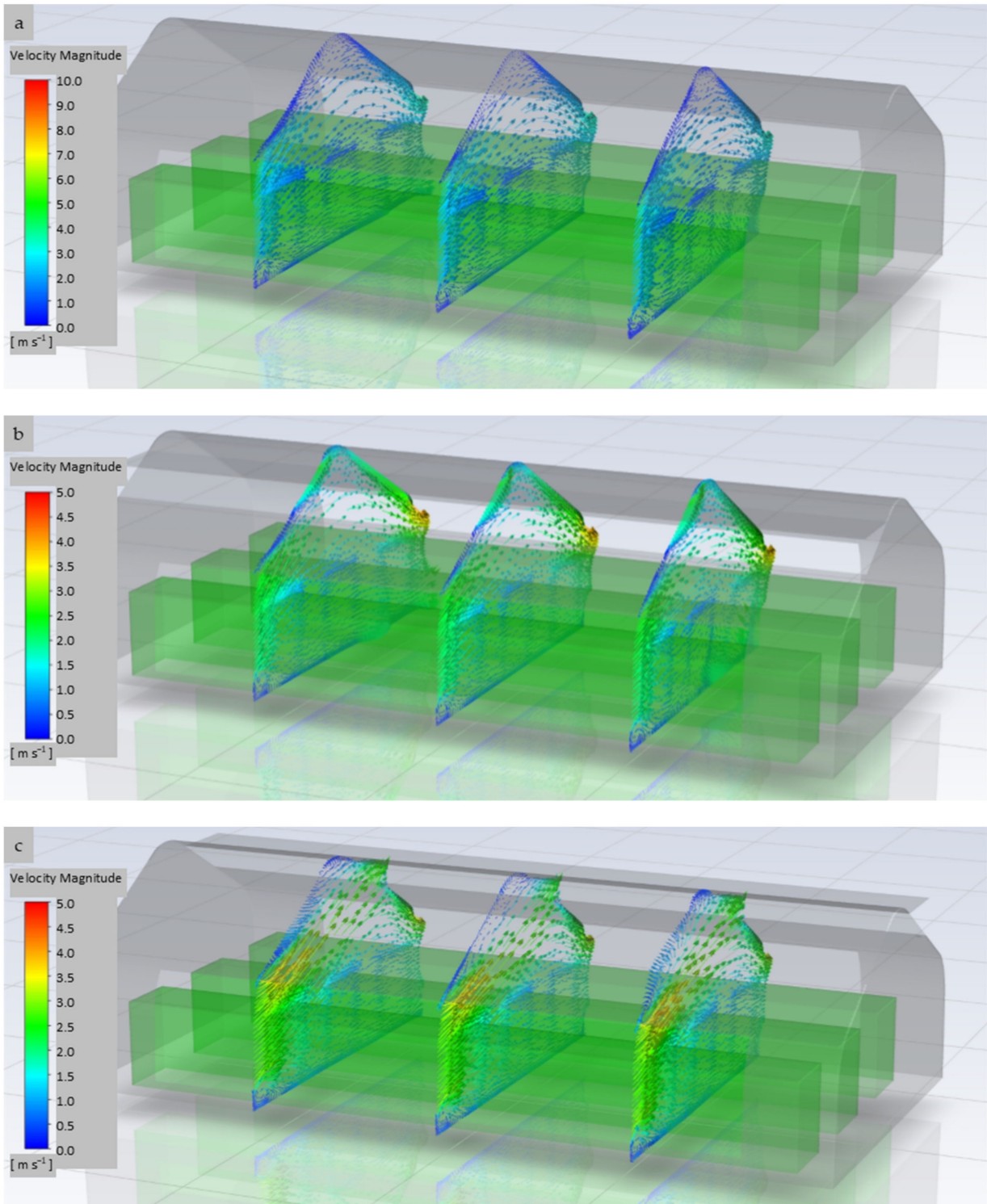

**Figure 9.** *Cont.*

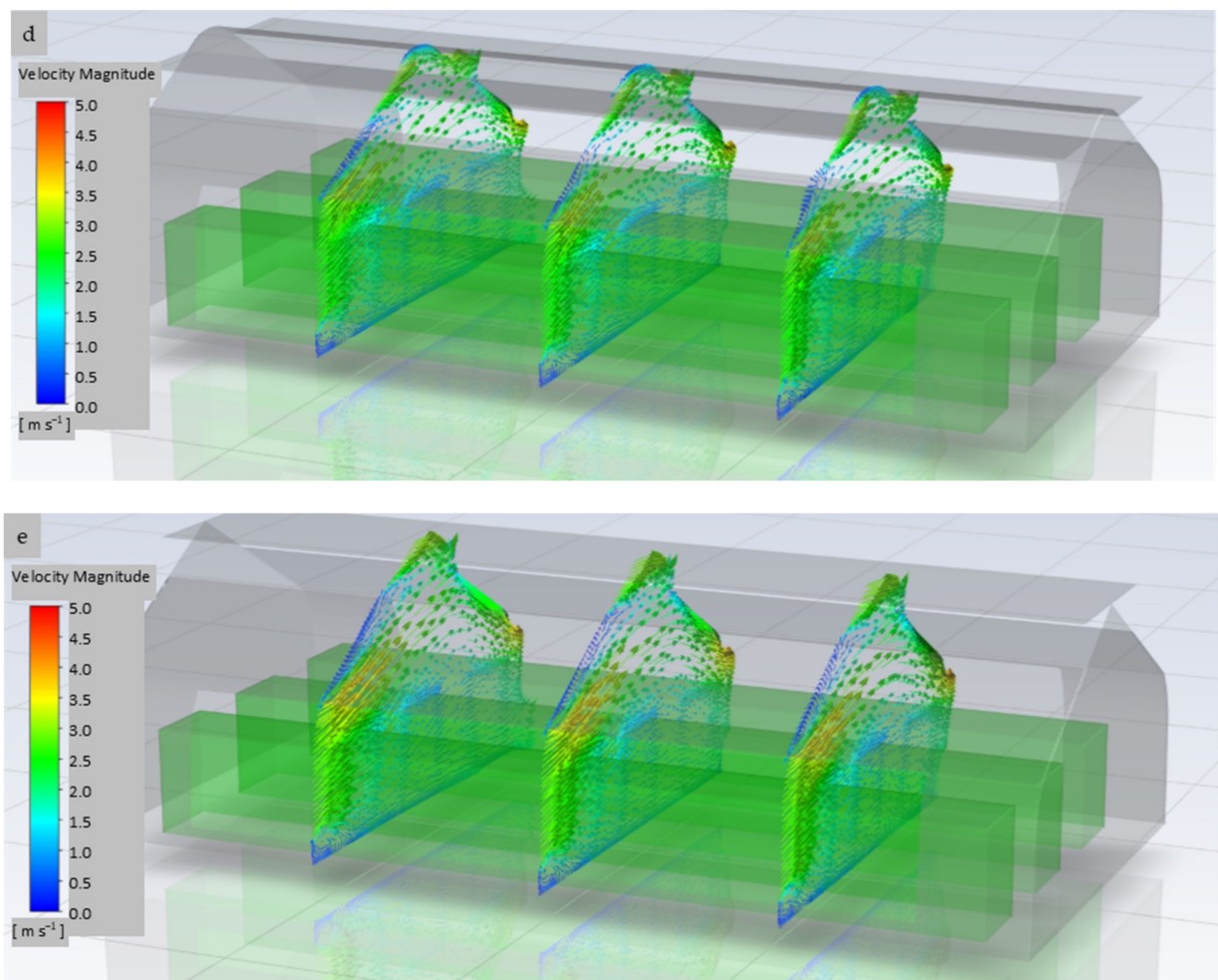

**Figure 9.** Simulated velocity vectors at three representative cross-sections within high tunnels outfitted with different roof vent designs: (**a**) roll-up side vents only; (**b**) added windward roof vent; (**c**) added leeward roof vent; (**d**) added windward and leeward roof vents; and (**e**) added ridge vents.

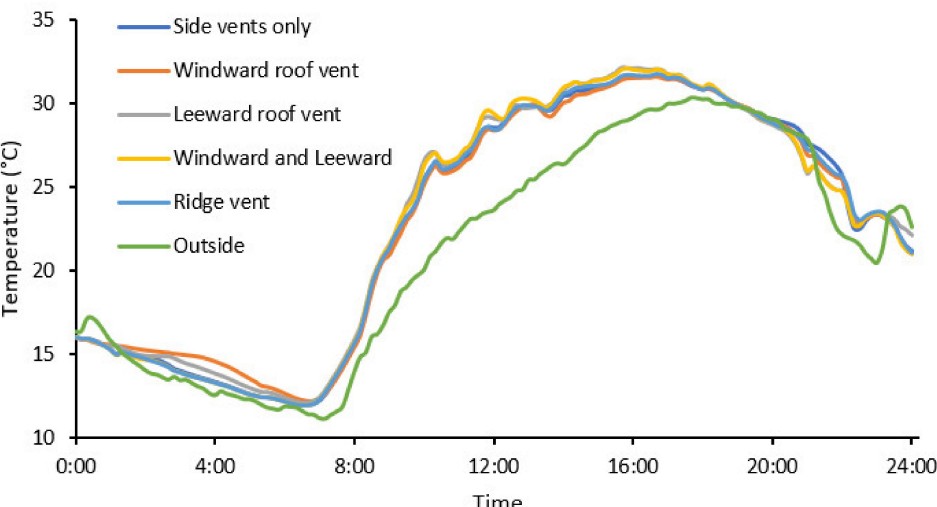

**Figure 10.** Simulated temperatures inside a high tunnel outfitted with different roof vent designs, averaged over the respective cross-sections depicted in Figure 9.

**Table 5.** Simulated average temperatures inside a high tunnel depending on the configuration of the roof vent. The temperature differential was calculated by subtracting the measured average outside temperature from the simulated average inside temperature. Average outside wind speed at 2.5 m: 1.5 m s$^{-1}$. Wind direction: 96.4° (relative to side vents). High tunnel dimensions: 11 m × 5.2 m × 3 m (length × width × height).

| Roof Vent Design | Average Inside Temperature, °C | Average Temperature Differential, °C |
|---|---|---|
| Roll-up side vents only | 22.9 | 1.5 |
| Added windward roof vent | 23.0 | 1.4 |
| Added leeward roof vent | 23.1 | 1.8 |
| Added windward and leeward roof vents | 23.0 | 1.7 |
| Added center ridge vents | 22.9 | 1.6 |

## 4. Discussion

This study investigated whether a freestanding high tunnel would benefit from the addition of a roof vent under summer conditions (i.e., the season with highest air temperature and solar radiation). The simulations used input parameters measured on a single day (13 July 2019) with optimal summer conditions, including close to no cloud cover, moderate wind from the prevailing wind direction, and a non-extreme air temperature. This approach limited the validation of the simulation model across a broader range of environmental conditions. The absence of cloud cover led to maximum radiative heat capture by the high tunnel and simplified the effects of cloud cover radiation interference/interaction in the simulation. Furthermore, on a day with less incident radiation, there would be less radiative heat capture, which would lower the ventilation demand. In addition, moderate wind from the prevailing wind direction allowed for acceptable performance (i.e., maintaining temperatures acceptable for raspberry cultivation) of a freestanding high tunnel with only roll-up side vents. The results obtained from this study confer with other similar work [9,11]. Namely, the combination of a roof vent and side vents leads to higher airflow through a high tunnel and similar single-span greenhouses.

It is unclear if any of the roof vent designs would perform significantly better (i.e., lower inside-outside temperature differential) than roll-up sides only if there was a lower wind speed or a wind direction that was not perpendicular to the roll-up side vents. The effect of vent configuration has been shown to impact temperatures inside single-span greenhouses when flow is primarily driven by buoyancy [11]. If similar conditions were applied to the vent designs used in this study, it can be assumed that there would be lower air flow rates overall. Nonetheless, future work needs to be done to determine at what point such conditions affect inside temperatures substantially. On the other hand, a much higher wind speed may show that some of the roof vent designs are impractical due to potential damage to the plastic film under extreme wind pressures. The moderate air temperatures simulated were suitable for raspberry cultivation at every point during the day.

Additional work is necessary to determine how the environment evolves over the course of a colder day. Most likely, a lower air flow rate would be acceptable to ventilate a high tunnel on colder days. Less air would need to be exchanged to maintain an optimal growing environment, since the air outside would be significantly colder than the air inside the high tunnel. However, under those conditions, greater consideration would need to be given to humidity control in order to prevent conditions conducive to humidity-induced plant diseases. On top of this, adding humidity and evapotranspiration to the model could allow for more accurate modeling of buoyancy and crop temperature. Overall, future simulations to assess additional environmental conditions are needed to fully understand the impact of roof vents on the growing environment inside freestanding high tunnels.

The practicality of various vent designs should be considered by manufacturers. There are physical limitations, such as fixing plastic to the tunnel. There are also cost considerations. The purpose of high tunnels is to minimize cost, and unnecessary modifications go

against this purpose. Future work should investigate what conditions would allow for the minimum and acceptable maximum temperature differential. While these are interesting scientific questions, it is important to consider their practical applications. Growers do not have control over the weather and only over site placement and high tunnel design. The described future work can help create recommendations for optimal site placement (i.e., a location with adequate wind) and design (i.e., vent designs in light of prevailing wind conditions).

## 5. Conclusions

Adequate ventilation through a high tunnel is critical to ensure a proper growing environment for crops. From a practical point of view, growers and manufacturers have limited options to improve ventilation given the low-cost philosophy associated with high tunnel crop production. This study simulated several roof vent designs using a 3-D CFD software package to investigate their impact on temperature and air flow rate. A transient simulation was used to examine the evolution of the internal high tunnel conditions over the course of a day. These simulations included the effects of radiative heat transfer and drag caused by the crop canopy.

Results were validated by comparing simulation results of one ventilation design (roll-up side vents only) with actual measurements collected in an experimental high tunnel with the same ventilation design. For the other ventilation designs investigated, no additional validations were conducted. The highest simulated ventilation rate was observed for a freestanding high tunnel outfitted with roll-up side vents and an additional ridge vent, with a 78% increase in the mass-based ventilation rate over the initial design, roll-up side vents only. However, the addition of any of the roof vent configurations investigated did not result in improved temperature control.

Additional research is needed to determine whether the addition of a roof vent is warranted in order to improve the environmental conditions inside freestanding high tunnels exposed to different outside conditions than the ones investigated for this study. During high temperature and low wind speed conditions, an increased mass-based ventilation rate caused by adding a roof vent may significantly lower the inside temperature of a high tunnel compared to a high tunnel outfitted with only side vents. These conditions were rarely experienced at the research site used in this study, limiting the ability to validate simulation results under such conditions. However, these conditions are common in other climates and further research could help show the benefits of high tunnel roof vents in those regions. In addition, further improvements to the model, such as accurate crop transpiration and a more detailed 3-D plant canopy representation, can improve the predictive power of the simulations.

**Author Contributions:** Conceptualization, D.C.L. and A.J.B.; methodology, D.C.L. and A.J.B.; software, D.C.L.; validation, D.C.L.; formal analysis, D.C.L.; investigation, D.C.L. and A.J.B.; resources, A.J.B.; data curation, D.C.L.; writing—original draft preparation, D.C.L.; writing—review and editing, D.C.L. and A.J.B.; visualization, D.C.L.; supervision, A.J.B.; project administration, D.C.L. and A.J.B.; funding acquisition, A.J.B. All authors have read and agreed to the published version of the manuscript.

**Funding:** This research was partially funded by the United States Department of Agriculture's National Institute of Food and Agriculture (Specialty Crops Research Initiative Agreement 2014-51181-22380; TunnelBerries project).

**Institutional Review Board Statement:** Not applicable.

**Data Availability Statement:** The data presented in this study are available on request from the corresponding author.

**Acknowledgments:** We gratefully acknowledge the support from Kathy Demchak and Matthew Cooper from Pennsylvania State University for managing the experimental high tunnel site; Claude Wallace for his help instrumenting the experimental high tunnel; and Thomas Bartzanas and Dimitris Chantzaras for their helpful suggestions regarding the CFD modeling.

**Conflicts of Interest:** The authors declare no conflict of interest. The funders had no role in the design of the study; in the collection, analyses, or interpretation of data; in the writing of the manuscript, nor in the decision to publish the results.

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
