# Peer review of "Using Computational Fluid Dynamics to Evaluate High Tunnel Roof Vent Designs"

_agriengineering, doi:10.3390/agriengineering4030046_

Round 1

Reviewer 1 Report

The manuscript deals with the analysis and the improvement of ventilation strategies in greenhouses using computational fluid dynamic simulations. In general, the topic falls within the aim and scope of Agriengineering Journal. In addition to that, the manuscript is well-structured, and the provided results might be interesting to the designers in the field. There is only one minor issue that the authors should pay attention and revise their work.

1.       Table 3. Please directly use the provided references in the table, instead of illustrated them as a footnote.  

Based on the above I can recommend this work for publishing after the successful manage of the above issue.

Author Response

Thank you for your helpful comments. Below are our replies to your comments and suggestions:

  1. We removed the footnotes for Table 3 and directly referenced our sources, except for the plastic cover reference. For that one, we can either leave the URL as a footnote or add a reference bracket [#] and move that link to our references.

Reviewer 2 Report

This study focuses on maximizing the air exchange rate results in a low differential between inside and outside air temperatures of five high tunnel ventilation during the summer, using CFD. The work is in the scope of the journal, however, redaction and structure should be improved as indicated below, especially the methods should be clearer; the author is recommended to identify and practice sophisticated objectives for a journal publication. The author must justify the following points:

Comment 1: Title should be revised. It would better to avoid using a direct question in the title of paper publication.

Comment 2: It was impossible for me to identify the novelty of the paper. The paper should be revised to highlight novelties. Please consider that this lack of novelty starts with the Abstract, Introduction, and Conclusion. Besides, a deep analysis of recent scientific papers covering only the topic and leading to the submission hypothesis based on the gap analysis of the previously published research is required.  

Comment 3: The proposed approach in section 2 is not outlined with the necessary vigor. The author needs to include sufficient methodological details in the paper and elaborate on the produced results from the proposed methods. Some sections must be added and others need to be relocated and rewritten to make it clearer for the readers. The sensors used to collect environmental data inside and outside the high tunnel must be explained. Subsection (2.2.) and (2.3.) must be better explained to justify the simulation of CFD and its limitations. How the input values presented in Table 2 were calculated? Figure 4 must be explained. How Figure 5 was simulated? How the values presented in Table 3 were estimated?

Comment 4: Figures 6, 7, 8, 9, and 10 must be better explained and justified. What are the applied materials and database to sort out such results?

Comment 5: More comparisons and Figures are required in the Discussion section to validate the simulated temperatures and velocity vectors at the three representative cross-sections within high tunnels out-fitted with different roof vent designs.

Comment 6: The Conclusion section is missing some necessary details. For example, the author needs to highlight the novelty and the materials and methods used in this work. Then the author should present the results of this work. Eventually, a summary of the limitations of this research as well as the recommendation for future works should be indicated.

Comment 7: Please consider that this is a scientific journal publication, where you need to avoid some phrases like (we, our, ….). Instead, you can use (this work, this study, this analysis….).

Author Response

Thank you for your helpful comments. Below are our replies to your comments and suggestions:

  1. We wanted to use this title to highlight the practical application of this paper. Here are some alternative titles we can suggest:
    • The effect of roof vent design on high tunnel ventilation
    • Assessing summer ventilation of high tunnels
  2. We added more references to our introduction to demonstrate the novelty of this paper. Ultimately our goal was to generate practical recommendations for high tunnel growers. We added text to highlight the novelty of the research in the abstract, and reiterated this in the introduction and conclusion. This is essentially that we simulated a 3D climate evolution model throughout the course of day to analyze the impact of roof vent designs on ventilation and ultimately temperature within a high tunnel. Our research is still limited and we can expand on these methods by simulating varying weather conditions. This is now stated in the abstract and conclusion.
  3. Since this comment addresses several points about our materials and methods section, we've broken our response into bullet points:
    • The accuracy and range of sensors used is added to Table 1. We believe it should be apparent to the reader why these sensors were used based off the information in section 2.3.
    • We don’t believe a significantly more explanation of the methods is necessary for sections 2.2 and 2.3. If readers want further understanding of the methods used, there are several references included that are detailed. We added more discussion of the limitations of our methods in these sections. Our introduction provides sufficient references to show how similar studies justify using CFD.
    • The values in Table 2 were not calculated, but measured at midnight on July 13, 2019. This was originally stated in the text, but is now also clarified in the table caption.
    • A brief justification of the designs presented in Figure 4 was added (line 158-159). The figure itself has a good explanation and visualization of what the designs are.
    • The data presented in Figure 5 was measured at our experimental site, not simulated. This is stated in the figure’s caption and in the text. We clarified that the reader can reference figure 1 to see where the sensors were that measured this data (line 163).
    • The material properties in Table 3 were gathered from several references. The reader can view the references for further information on how these properties were estimated.
  4. We added more discussion for each of these figures to better explain and justify them. In our methods, we added a mention that the results were compiled and analyzed using Microsoft Excel.
  5. We didn’t validate the velocity vectors in the tunnel because of limited wind data, so the velocity vector results are simulated only. Further work is needed to validate this. However, our temperature was validated in figure 6 against the side vent only design. This took into account 3 cross sections of the tunnel to create an average temperature, which was compared to the average temperature of our measured data. We do not claim that our simulations have the power to accurately predict localized temperatures within the high tunnel, which would require us to validate air velocities, use a higher mesh count, and apply more detailed physics. Ultimately, this was not the focus of our study.
  6. We added the recommended details to our conclusion.
  7. We changed the phrasing throughout to remove first person grammar.

Reviewer 3 Report

Comments on the manuscript: “Do roof vents improve high tunnel ventilation?”. The English language is very good, no jargon is detected. The idea of analyzing the effect of roof vents on ventilation is good. However, I believe that the results are in the initial stage, and they are very limited. Unless the authors address all points mentioned as future work. I hope the following comments can help the authors.

If the question presented as the title of the manuscript is not answered, what do the authors believe a reader would find after reading your manuscript? Nothing, it just concludes that further work is needed.

No quantitative results are shown in the abstract.

The introduction needs to provide enough background to place the research into context and give readers a good understanding of what the paper is about. It should be clear what is already known on this topic by covering an adequate literature review. It should identify the patterns or trends and highlight the most recent findings within this topic, with a clear gap in knowledge that the current paper seeks to address. The research questions or hypothesis should be clearly stated. 

I believe that in general sections 1 and 2 are correct for a manuscript, but section 3 is incomplete.  

Figure 1 is not labeled correctly; which of the three figures is Fig 1(b) and which is Fig 1(c)?

Figures 5(b) and 5(c) could be joined into a single figure as Fig 5(a)

What is the maximum percentage difference between experimental and measured temperatures?

I believe that lines 167-171 belong to section 2.4.

Please explain to the reader how did you notice that more air passes over the top of the plants? Extend your explanation.

About comment in lines 185-186, Compared to which case causes less air to pass through the plants?

Author Response

Thank you for your helpful comments. Below are our replies to your comments and suggestions:

  1. If the question presented as the title of the manuscript is not answered, what do the authors believe a reader would find after reading your manuscript? Nothing, it just concludes that further work is needed.
    • We wanted to use this title to highlight the practical application of this paper. This question is directly answered in the abstract (lines 19-21) and conclusion (lines 300-304)
    • Here are some alternative titles we can suggest:
      • The effect of roof vent design on high tunnel ventilation
      • Assessing summer ventilation of high tunnels
  2. No quantitative results are shown in the abstract.
    • We added the root mean square error of our validation results, as well as the qualitative results of mass based ventilation rates and temperature.
  3. The introduction needs to provide enough background to place the research into context and give readers a good understanding of what the paper is about. It should be clear what is already known on this topic by covering an adequate literature review. It should identify the patterns or trends and highlight the most recent findings within this topic, with a clear gap in knowledge that the current paper seeks to address. The research questions or hypothesis should be clearly stated.
    • We added more references to our introduction to give a deeper context. They show the trend towards transient simulation studies, with a clear lack of 3-D studies and a lack of studies focused on improving greenhouse design (majority focus on understanding the system better or examining microclimates)
    • We added a clear statement of our research question (do roof vents improve high tunnel ventilation) at the end of the introduction.
  4. I believe that in general sections 1 and 2 are correct for a manuscript, but section 3 is incomplete.
    • We’re not sure how to address this directly, but based off the feedback from other reviewers we’ve added clarifications and further discussion in section 3.
  5. Figure 1 is not labeled correctly; which of the three figures is Fig 1(b) and which is Fig 1(c)?
    • This was a formatting issue and has been fixed.
  6. Figures 5(b) and 5(c) could be joined into a single figure as Fig 5(a)
    • We believe this will be confusing, the current style clearly distinguishes the 3 separate climate measurements.
  7. What is the maximum percentage difference between experimental and measured temperatures?
    • We include this and the minimum error, as well as explanation of these differences (lines 190-197)
  8. I believe that lines 167-171 belong to section 2.4.
    • These lines include the results of our validation methods, so we believe they belong in the results section.
  9. Please explain to the reader how did you notice that more air passes over the top of the plants? Extend your explanation.
    • An extended explanation was added for Figure 9.
  10. About comment in lines 185-186, Compared to which case causes less air to pass through the plants?
    • This comparison is to the two preceding designs (side vent only and windward roof vent). This was clarified.

Round 2

Reviewer 2 Report

The work has developed and the author answered my previous comments. There is a need to change the title to be more suitable for the presented manuscript, please. 

Author Response

Thank you again for your helpful comments. We have changed the title of the manuscript to the following: "Using computational fluid dynamics to evaluate high tunnel roof vent designs".

Reviewer 3 Report

The authors have attended most of the comments raised by the reviewer, I have some minor comments on the manuscript:

In line 16 of the abstract replace " simneulated" by "simulated"   

In the discussion, the results obtained by the authors should be compared to the available data noted in the literature review to highlight any new contribution of the work presented in the paper.

Author Response

Thank you again for your helpful comments. We have fixed the typo mentioned in line 16 and added more discussion (lines 261-263 and 266-270) to compare our work to our references.